# Human Colonoid–Myofibroblast Coculture for Study of Apical Na^+^/H^+^ Exchangers of the Lower Cryptal Neck Region

**DOI:** 10.3390/ijms24054266

**Published:** 2023-02-21

**Authors:** Azam Salari, Kunyan Zhou, Katerina Nikolovska, Ursula Seidler, Mahdi Amiri

**Affiliations:** 1Department of Gastroenterology, Hepatology and Endocrinology, Hannover Medical School, 30625 Hannover, Germany; 2Department of Thyroid Surgery, The First Affiliated Hospital, School of Medicine, Zhejiang University, Hangzhou 310027, China

**Keywords:** human intestinal organoids, colonic electrolyte transport, sodium–hydrogen exchange, SLC9 family, intestinal barrier, claudins, pH_i_ regulation

## Abstract

Cation and anion transport in the colonocyte apical membrane is highly spatially organized along the cryptal axis. Because of lack of experimental accessibility, information about the functionality of ion transporters in the colonocyte apical membrane in the lower part of the crypt is scarce. The aim of this study was to establish an in vitro model of the colonic lower crypt compartment, which expresses the transit amplifying/progenitor (TA/PE) cells, with accessibility of the apical membrane for functional study of lower crypt-expressed Na^+^/H^+^ exchangers (NHEs). Colonic crypts and myofibroblasts were isolated from human transverse colonic biopsies, expanded as three-dimensional (3D) colonoids and myofibroblast monolayers, and characterized. Filter-grown colonic myofibroblast–colonic epithelial cell (CM-CE) cocultures (myofibroblasts on the bottom of the transwell and colonocytes on the filter) were established. The expression pattern for ion transport/junctional/stem cell markers of the CM-CE monolayers was compared with that of nondifferentiated (EM) and differentiated (DM) colonoid monolayers. Fluorometric pH_i_ measurements were performed to characterize apical NHEs. CM-CE cocultures displayed a rapid increase in transepithelial electrical resistance (TEER), paralleled by downregulation of claudin-2. They maintained proliferative activity and an expression pattern resembling TA/PE cells. The CM-CE monolayers displayed high apical Na^+^/H^+^ exchange activity, mediated to >80% by NHE2. Human colonoid–myofibroblast cocultures allow the study of ion transporters that are expressed in the apical membrane of the nondifferentiated colonocytes of the cryptal neck region. The NHE2 isoform is the predominant apical Na^+^/H^+^ exchanger in this epithelial compartment.

## 1. Introduction

Based on the study of gene-deleted mice, the functions of intestinal Na^+^/H^+^ exchangers (NHEs) are essential for all aspects of mucosal physiology, including proliferation and differentiation, host defense, mucus barrier properties, wound healing and, of course, fluid and electrolyte transport [1,2,3]. Because the NHE-deficient mice develop colonic barrier impairment [4], dysbiosis [5], or a disturbed differentiation pattern [6], it has been difficult to define the cellular physiology of each transporter. Except for NHE3, immunohistochemical localization of these NHE isoforms is controversial, and functional data in the human colon are missing.

Recently, intestinal epithelial proliferation and differentiation have been shown to be critically linked to the Na^+^/H^+^ exchangers NHE2 [6,7] as well as NHE8 [8,9] in mice and in the Caco-2BBe intestinal cell line. Data from genome-wide association studies (GWAS) in humans also found polymorphism in the *SLC9A8* gene, which encodes NHE8, to be critically linked to cancer progression [10]. The data suggested that the two isoforms have an important but fundamentally different role in cellular physiology, which requires further investigation. Although intestinal cell lines have been successfully used to characterize multiple aspects of NHE physiology, their origin from malignant tumors is a reason for concern. Therefore, experimental systems that allow the functional study of ion transporters in native human colonocytes are necessary.

Elucidation of the relevant factors and development of protocols for maintenance of intestinal epithelial stem cells in vitro enabled scientists to grow, passage and differentiate human intestinal organoids derived from biopsy material [11,12]. The crypts isolated from human colon form three-dimensional “colonoids” with the apical membranes facing an inner lumen, and cryptal buddings, which contain the proliferating cells. These colonoids display a differentiation gradient from the base of the crypts to the cells facing the inner lumen [13].

For the functional characterization of acid–base transporters, complex three-dimensional structures with different cell types, such as the intestinal epithelium, have methodological disadvantages because the apical membranes of the cells in the cryptal region are not accessible. The same is true for three-dimensional organoids. In this study, therefore, we first developed a protocol to culture human three-dimensional colonoids from healthy subjects in a stem cell-enriched state and a differentiated state and studied the expression changes of a panel of acid–base transport proteins during the differentiation process. We then used this information to model the organoids as monolayer cultures in different developmental states: (i) nondifferentiated, low resistance monolayers resembling the cells in the colonic stem cell niche (EM); (ii) a novel epithelial platform consisting of *KI67* positive, nondifferentiated, high resistance epithelial–myofibroblast cocultures (CM-CE), resembling the transit amplifying/progenitor (TA/PE) cells in the lower cryptal region, and (iii) differentiated cells resembling the colonic epithelial cells in the upper cryptal region (DM). This study then focused on further characterization of the CM-CE monolayers and studied their apical NHE isoforms by a fluorometric technique. This allowed, for the first time, a determination of the NHE isoform that governs apical NHE activity in the lower cryptal region of the human colon.

## 2. Results

### 2.1. Morphological Features and Expression Profile of Differentiation Markers and Ion Transporters in Nondifferentiated and Differentiated 3D Human Colonoids

A protocol was established to generate, expand and differentiate human colonoids from the transverse colon of healthy volunteers. The protocol resulted in qualitatively similar results between donors. Organoid cultures gave similar functional results when expanded, frozen, thawed and expanded again. Figure 1A displays the morphological features of nondifferentiated and differentiated three-dimensional (3D) organoids (the differentiated organoids were cultured for 4 days in differentiation medium (DM), while the nondifferentiated organoids remained in the expansion medium (EM). Immunohistochemical staining of whole mount organoids (Figure 1B) shows abundance of KI67 positive proliferating cells in the expansion medium, while the differentiated organoids show hardly any proliferating cells, but stain for differentiation markers such as the goblet cell marker mucin-2 (MUC2) and absorptive enterocyte markers NHE3 and the apical Cl^−^/HCO_3_^−^ exchanger SLC26A3.

The log_10_-fold change in the expression levels of differentiation markers and ion transporters in Figure 2A displays the dramatic changes in expression levels as the organoids undergo differentiation. These changes correspond with the expression changes of a variety of transporters along the cryptal axis in murine or human colon [14,15,16,17,18,19,20]. For selected ion transporters, this was also observed by western blot analysis (Figure 2B). Nevertheless, the absolute mRNA and protein expression levels do not reach those of the native colonic epithelium (Figure 2C,D and Appendix A). The relative expression pattern of the different apical ion transporters is preserved, however.

### 2.2. Modeling of Two-Dimensional Organoid Monolayers with Myofibroblast Coculture

Nondifferentiated and differentiated organoids have previously been functionally studied in monolayer culture, for example from the duodenum [21], the proximal colon [22] and the rectum [23]. The described protocols yield monolayers with low resistance if grown in expansion medium and high resistance if grown in differentiation medium [21,22,23]. In fact, the development of an electrical resistance is often used as a “differentiation” marker [24]. However, it is known that human colonic cell lines such as the T84 cells display very high electrical resistance yet are models for the chloride secretory cells in the lower part of the crypts [25]. In contrary, the Caco-2BBe cells represent a well-organized brush border membrane and express a high level of absorptive electrolyte transporters such as SLC26A3 [22] yet display a much lower electrical resistance than T84 cells. A variety of different culture conditions were tested, including epithelial–myofibroblast coculture. Pericryptal myofibroblasts are known to secrete a large variety of bioactive substances [26] and have been described to maintain stem cell function [27,28,29], improve crypt formation and differentiation [30,31]. A subpopulation of colonic fibroblasts which is located at epithelial stem cell niche can be distinguished by CD90 expression [29]. We therefore isolated and cultured myofibroblasts from the lamina propria of human transverse colon. The majority of these myofibroblasts stained positive for CD90 in immunofluorescent analysis (Figure 3 and Appendix A). After confluency, the colonoid monolayers on transwell filters were then placed into the culture dish that had a myofibroblast monolayer at the bottom of the dish (Figure 4A). This resulted in a rapid increase in TEER (Figure 4B), and the formation of an optically homogeneous monolayer (Figure 4C). The immunohistochemistry and expression profile of the cells in the bottom of the culture dish (myofibroblasts) was consistent with a myofibroblast-enriched monolayer secreting a variety of growth factors (Figure 3A,B). The expression profile of the organoid monolayer was consistent with a cell layer resembling transit amplifying/progenitor (TA/PE) cells in the lower part of the colonic crypt, with much lower *LGR5* than the stem cell-enriched (SE) cultures, but comparable *KI67* (Figure 4 and Figure 5) to EM monolayers. CM-CE culture further shows decreased *NKCC1* and *NBCn1* compared to SE and EM organoids, and no increase in *SLC26A3*, *NHE3*, *ALPI* or *MUC2*, which was seen in the differentiated enterocytes/goblet cell (DE/DG) monolayers (Figure 5).

### 2.3. Expression Changes in Tight Junctional Components during Monolayer Differentiation

An increase in electrical resistance of an organoid-derived monolayer has been defined as a marker for differentiation [21,23,32,33]. However, the placement of the nondifferentiated monolayers onto the myofibroblast layer while maintaining the “expansion medium” resulted in a similar time pattern of TEER increase to that described for the generation of differentiated colonoid monolayers by withdrawal of Wnt-3a and R-spondin [21,23], yet no increase in the differentiation markers *ALPI*, *SLC26A3* or *MUC2* (Figure 5). To investigate potential molecular mechanisms causally related to the rapid TEER increase, we assayed the expression changes in a panel of tight junctional components (Figure 6A). While the mRNA expression of *claudin-1* and *-8* did not substantially change during differentiation, that of *claudin-3,-4,-7*, *ZO-1* and *occludin* was upregulated during differentiation, and that of *claudin-2* was strongly downregulated. The only dramatic difference between EM and CM-CE organoids was the strong decrease in the expression of the pore-forming *claudin-2*. Immunohistochemical staining of claudin-2, ZO-1 and cytoskeletal proteins confirmed the strong downregulation of claudin-2 from the 3D SE organoids to 2D (Figure 6B). In contrast, the upregulation of the tightening claudins was observed only in the DM monolayers (Figure 6A). Since claudin-2 deletion results in an increase, and claudin-2 expression results in a decrease of TEER in the intestine and in cell lines [34,35,36,37], one likely molecular reason for the rapid increase in TEER in CM-CE is the sharp decrease in claudin-2 expression, which results in a decrease in paracellular cation conductance.

### 2.4. Expression and Fluorometric Characterization of Apical Na^+^/H^+^ (NHE) Exchange in CM-CE Monolayers

Our previous study on NHE expression and function in murine distal colon revealed an important function of NHE2 in the differentiation of the colonocytes via regulation of the intracellular pH of the transit amplifying cells [6], where it was shown to be indispensable by any other NHE isoform. To analyze whether the currently presented CM-CE monolayers as a model for transit amplifying cells resembles the findings in the murine colon, we assessed the mRNA expression levels for the different intestinally expressed *NHE*s (Figure 7A). Expression of the *NHE* isoforms *NHE2* and *NHE8* does not change with differentiation, confirming the results in the three-dimensional organoids (Figure 2A), while expression of *NHE3* increases strongly and expression of *NHE1* increases slightly with differentiation. In contrast, the base loader *NBCn1* is highly expressed in SE organoids and is downregulated during differentiation (Figure 5).

The mRNA expression levels of *NHE* isoforms need not parallel functional activity, because heterologously expressed NHE1-3 proteins have been shown to have very different lifetimes [38]. We assessed their functional activity by BCECF fluorometry (Figure 7B–I). As shown previously for Caco-2BBe cells, pH_i_ recovery from an ammonium prepulse after selective addition of Na^+^ to the basolateral perfusate was rapid (Figure 7B,C), and is mediated by NHE1 [9,11]. To access the apical exchange activity, the monolayer cultures were perfused from the apical side with Na^+^-containing perfusate (after an ammonium prepulse) in the presence of different concentrations of the inhibitor HOE642 to selectively inhibit each NHE isoform (described in [7,9]). To eliminate a potential contribution of basolateral NHE1 to the apical NHE activity (Figure 7D–G), Na^+^ ions were removed from the basolateral perfusate, but 3 µM HOE642 was added to the basolateral perfusate to inhibit NHE1 activity potentially driven by apical Na^+^ ions penetrating through the tight junctions. Apical Na^+^-dependent pH_i_ recovery from an ammonium prepulse was then assessed in the presence of apical 3 µM HOE642, which fully inhibits human NHE8 [9] (Figure 7E). The residual recovery rate (ΔpH/min) observed in this case (Figure 7H,I second column) is a sum of NHE2 and NHE3 activity. Addition of 1 µM tenapanor to 3 µM HOE642 to fully inhibit NHE8 and NHE3 (Figure 7F) did not further decrease apical Na^+^-dependent pH_i_ recovery (Figure 7H,I, first three columns). However, when the apical Na^+^-dependent pH_i_ recovery was measured in the presence of 60 µM HOE642, which fully inhibits NHE8 and NHE2 (Figure 7G), a substantial reduction in the Na^+^-dependent pH_i_ recovery rate was observed (Figure 7H,I, first three columns vs. fourth column). The relative NHE-specific transport rates were calculated by setting the apical Na^+^-dependent pH_i_ recovery rates in the absence of inhibitors to 100% and calculating the rates in the presence of inhibitors as percentage values (Figure 7I).

## 3. Discussion

Human and rodent colons are able to absorb large amounts of electrolytes and fluid with a smaller membrane area than the small intestine. The hypothesis established in the 1970s and 1980s was that in the intestine, the surface/villous cells absorb electrolytes and water, and the cryptal cells secrete anions and water [39,40,41]. The ion transporters that mediate these electrolyte and fluid movements were characterized both in intestinal perfusion studies in vivo, and in isolated membrane vesicles in vitro, and a parallel operation of Na^+^/H^+^ exchange and Cl^−^/HCO_3_^−^ exchange in the brush border membrane of the villous/colonic surface cells was postulated [42,43,44,45,46,47].

Nevertheless, the hypothesis of an exclusively surface cell-located absorptive capacity was experimentally challenged. In 1995, Singh et al. described that active fluid absorption was observed in isolated, luminally perfused rat colonic crypts, potentially mediated by cryptal Cl^−^-dependent Na^+^/H^+^ exchange [48,49]. Simultaneously, the different members of the Na^+^/H^+^ exchanger family were cloned, and an intestinal expression was demonstrated for NHE1-4 and NHE8 [50,51,52]. While a major role for the surface cell-expressed NHE3 isoform for intestinal fluid absorption was ascertained [53,54], the role of the other apical isoforms was less clear. Recent progress on a better understanding of the apical NHEs in the intestine is summarized in recent reviews [6,55].

This study describes the establishment of filter-based colonoid–myofibroblast cocultures from the human transverse colon. This was performed to model the lower cryptal neck region which predominantly contains the nondifferentiated transit amplifying/progenitor cells in vitro, and to be able to experimentally access their apical membrane for the functional identification of the apical Na^+^/H^+^ exchanger isoforms that are operative in that membrane. The high electrical resistance of this coculture system, which was likely predominantly due to the rapid downregulation of the pore-forming claudin-2 after exposing the nondifferentiated, low resistance filter-grown monolayers to the myofibroblast-conditioned medium, enabled the selective measurement of NHE2 and NHE8 activities, while completely inhibiting basolateral NHE1 activity. The discussion focuses on two different issues. Firstly, the establishment of the coculture system is discussed. Secondly, the methodological approach and significance of establishing the NHE isoforms active in the apical membrane of these CM-CE cultures are discussed. Finally, an outlook is given on the future studies that this coculture model allows to be undertaken.

To establish a suitable model for the functional study of the human colonic apical NHE isoforms, we followed established protocols for the generation, maintenance and differentiation of colonoids in 3D and 2D, with similar results to those described [12,22,23]. We noticed that in the filter-grown colonoids, TEER slowly increases, while the expression of the ion transporters that are highly expressed in the highly proliferative/stem cells gradually decrease (Figure 4 and Figure 5). When terminal differentiation is initiated by removal of Wnt stimulating factors in a period of 4 days, a very sharp increase in resistance, a virtually complete disappearance of *LGR5* expression and *KI67* positivity, a wavier surface of the monolayer, an increased apoptotic rate but also loosening of the cells from the matrix, was observed (Figure 4). Similar results have been obtained by others in rectal or duodenal organoids [21,23]. The low resistance of the nondifferentiated colonoids and the low matrix adherence of the terminally differentiated colonoids did not provide optimal conditions for functional NHE measurements with the clear target of separating apical and basolateral NHE activities, and we did not pursue NHE measurements in EM and DM monolayers.

Altay et al. cultured murine small intestinal organoid monolayers in medium enriched with subepithelial myofibroblasts (ISEMFs), because these cells represent an important alternative non-epithelial source of Wnt pathway activators [56], and observed an enhanced cell spreading and barrier function [57]. We, therefore, isolated and cultured myofibroblasts obtained from the lamina propria of human transverse colon and established the CM-CE coculture system described above. We observed a rapid increase in the TEER when the confluent colonoid cultures were placed on top of the myofibroblast monolayers seeded on the bottom of the culture dish. In a search for a molecular explanation for the rapid TEER increase, we assayed a panel of genes that encode for the paracellular barrier, as well as genes responsible for transcellular ion flux. The gene that was conspicuously downregulated between the EM and the CM-CE monolayer cultures was *CLDN2*, the gene for the pore-forming tight junction protein claudin-2. Claudin-2 has been characterized as a cation-selective conductance and a water pore [58], and its location in the colonic lower cryptal area and base suggests that it functions to bring Na^+^ and water to the lumen when anions leave via CFTR. Differences in claudin-2 expression have been shown to be responsible for dramatic differences in TEER within the same cell type [59]. Other genes that indicate proliferative activity (*LGR5*, *KI67*) were expressed to a similar degree as in EM cultures (but more downregulated than in the stem cell rich, highly proliferative SE 3D cultures), but the genes that indicated absorptive enterocyte or goblet cell differentiation did not increase as compared to EM cultures (Figure 5). This suggested that the CM-CE monolayers represented predominantly transit amplifying/progenitor cells.

We considered the CM-CE coculture an excellent model to fluorometrically characterize the apical Na^+^/H^+^ exchangers in the transient proliferating cells of the human colonic lower cryptal neck zone. In differentiated Caco-2BBe cells, basolateral Na^+^/H^+^ exchange needs to be fully inhibited to uncover the lower rates of apical Na^+^/H^+^ exchange. This requires a separate and rapid perfusion of the luminal and basolateral bath, the removal of Na^+^ from the basolateral bath plus an addition of a NHE1-selective concentration of HOE642 to the basolateral bath, to prevent a partial NHE1 activation via Na^+^ ions that permeate from the luminal compartment via the cation-selective tight junctions [9]. This uncovered a highly HOE642-sensitive Na^+^/H^+^ exchange activity in the apical membrane of fully differentiated Caco-2BBe cells, suggested to be NHE8 based on the inhibitor profile. In order to verify that this activity was due to NHE8, and not an artifact, we expressed flag-tagged NHE8 in these cells and demonstrated apical staining with an antibody against flag [9]. NHE2 and NHE3 were also active in the apical membrane of differentiated Caco-2BBe cells [8,11].

Using the same methodology, ~20% of the apical Na^+^/H^+^ exchange in the apical membrane of the CM-CE monolayers was inhibited by 3 µM HOE642. The NHE3-specific inhibitor tenapanor had no additional effect, consistent with the low expression of NHE3 in the CM-CE cocultures. However, 60 µM HOE642 in the apical bath, which inhibits NHE8 and NHE2, fully inhibited apical pH_i_ recovery. Because 3 µM HOE42 will also slightly affect NHE2 [60,61], this approach will overestimate apical NHE8 activity, which will more likely be ~10% of total apical NHE activity. Given its described role in intestinal proliferation [8], the low NHE8-attributable NHE activity surprised us. In some cell lines NHE8 has been reported as an organellar NHE that performs functions important for transporting cargo to the plasma membrane [62,63,64]. Immunohistochemical staining of the CM-CE monolayers with a previously validated anti-NHE8 antibody [9] demonstrated low intensity of subapical immunoreactivity (Appendix A), consistent with its low mRNA expression levels in the CM-CE cocultures, none was found in the apical membrane. Because of the low NHE8 activity even during acid-activation, and low *NHE8* mRNA and protein expression levels, the CM-CE coculture system does not seem a useful model to characterize intestinal NHE8 any further.

In contrast, the NHE2 isoform was found to be the predominant apical NHE isoform in the CM-CE cocultures, which correspond to the transit amplifying cells of the human colonic crypts. Similar results have previously been obtained in murine colonic crypts [65,66]. What may be the physiological necessity of a Na^+^/H^+^ exchanger in the luminal membrane of the colonocytes in the lower cryptal region, cells that are involved in fluid secretion rather than absorption, and that obviously have robust basolateral NHE1 exchange? Several features of NHE2, as well as of the colonocyte, come to mind. Firstly, NHE2 has a particularly high proton affinity to both the external and internal binding site, which allows NHE2 to be active in a pH range in which NHE1 is already quiescent [67,68]. We previously found that NHE2 was involved in absorptive colonocyte proliferation and differentiation in the enterocyte cell line Caco-2BBe as well as in the murine colonic epithelium, and that the molecular mechanism was likely an effect of NHE2 on steady-state pH of the enterocytes in the transit amplifying zone [6].

Secondly, the colonocytes, including those in the stem cell niche, are very long, thin cells, with a long distance between the basal and apical pole. Apical CFTR activity in the cryptal base and neck cells results in a Cl^−^, volume, and HCO_3_^−^ loss of these cells [69,70]. It is possible, even likely, that these ionic and volume alterations occur primarily near the apical cell pole. NHE2 may not only serve to regulate pH_i_ but also to regulate volume in the apical cell pole. An advantage of NHE2 over NHE3 (minimally expressed in the transit amplifying cells anyway) to counteract a secretion-associated volume- and base loss is that NHE3 activity is inhibited by cell shrinkage, whereas NHE2 is activated [71]. Furthermore, it has recently been shown that differentiated enterocytes in the gut actively migrate up the crypt-villus/surface axis while fully maintaining their apicobasal polarity and likely their absorptive function, suggesting that the apical and basolateral acid/base transporters may be engaged with different tasks, and that pH_i_ regulation at the apical and basal pole of the enterocyte follows different demands and is mediated by different transporters [72]. Whether native human colonocyte cellular differentiation is similarly modulated by NHE2-regulated pH_i_ than observed in the mouse colon and in Caco-2BBe cells awaits confirmation. Since the differentiation induced in colonoids is currently different from that in vivo, further optimization of the colonoid culture system may help better understand the modulation of differentiation by ionic conditions.

In summary, this study introduces and characterizes a novel colonic myofibroblast– enterocyte coculture model that allows functional study of the transit amplifying cells in the lower cryptal neck region. It confirms, for the human colonic epithelium, data obtained in the mouse colon which show that the Na^+^/H^+^ exchanger isoform 2 is the predominant apical NHE in the cryptal region, likely corresponding to the Cl^−^-dependent Na^+^/H^+^ exchanger first described by Rajendran et al. [62] with the Cl^−^-dependency being explained by the strong effects of luminal Cl^−^-removal on enterocyte pH, volume, and WNK (with no lysine kinase) pathway-operated transporters. For the functional study of the human colonic cryptal neck cells, this model appears optimal. A high epithelial resistance is combined with the cylindrical shape of human enterocytes, an absence of differentiation markers allows the study of growth, differentiation, barrier properties and anion secretory functions of human colonic cryptal cells. It may also serve to better understand lamina propria-epithelium interactions such as the effect of GLP2 agonists on intestinal epithelial growth and function.

## 4. Material and Methods

### 4.1. Human Colonoid and Myofibroblast Cultures

Human colon 3D organoid cultures were established from biopsies of healthy donors after institutional review board approval number 8536_BO_K_2019 from 26.06.2019 as already published in Zhou et al. [9]. Colonoids after two-day culture in expansion medium (EM) supplemented with 10 µM of Y-27632 (Tocris Bioscience, Wiesbaden-Nordenstadt, Germany), 10 µM of CHIR-99021 (Sigma-Aldrich Chemie GmbH, Taufkirchen, Germany) and 1 mM of valproic acid (pharmaceutical secondary standard) are referred to as “stem cell-enriched” (SE) cultures. To prepare differentiation medium (DM), L-WRN conditioned medium, SB202190 and PGE2 were excluded, and 10% (*v*/*v*) Noggin conditioned medium was included in the EM composition, which is described in detail in Zhou et al. [9].

Myofibroblasts were isolated from epithelium-free biopsies of transverse colon according to Khalil et al. [73]. Myofibroblasts were expanded in EMEM (Lonza, Basel, Switzerland) containing 10% FBS, 1% non-essential amino acids (Gibco, Karlsruhe, Germany) and penicillin/streptomycin (Sigma-Aldrich Chemie GmbH, Taufkirchen, Germany) at 37 °C with 5% CO_2_ and used between passages 6 to 12. Myofibroblasts with a density of 5.5 × 10^4^ cells/well were seeded in a 24-well plate, 2 days before starting the coculture.

For colonoid monolayers, transwell inserts (6.5 mm diameter polyester membrane with 3 µm pores; Corning GmbH, Kaiserslautern, Germany) were coated with 10 mg/cm^2^ human collagen IV solution (Sigma-Aldrich Chemie GmbH, Taufkirchen, Germany) and incubated at 37 °C for 2 h. The remaining solution was aspirated and air dried for 15 min under sterile bank, then washed three times with advanced DMEM F12 medium. EM supplemented with Y-27632 and CHIR-99021 was added to the inner insert chamber and the bottom of the well (100 and 600 µL, respectively) and incubated in the tissue culture incubator 30 min prior to adding the colonoid fragments. Three-dimensional colonoids were harvested and washed in cold DPBS, exposed to 0.02% trypsin-EDTA for 3 min at 37 °C and then triturated with a P200 pipet tip 2–3 times to obtain small fragments each with about 30 cells. Fragments were washed with advanced DMEM F12 medium containing 10% FBS, spun down at 200 g, 4 °C for 5 min and resuspended in 100 µL EM supplemented with Y-27632 and CHIR-99021, and then added to the upper chamber of the transwell insert. Formation of a confluent colonoid monolayer which was characterized as transepithelial electrical resistance (TEER) values higher than 150 Ω/cm^2^ was monitored by EVOM2 volt-ohm meter (World Precision Instruments, Sarasota, FL, USA). The transwell inserts with the confluent colonoid monolayers were transferred to the 24-well plate containing myofibroblast cultures on the bottom of the wells. Cells in the upper and the lower chambers were treated with EM during coculture according to the protocol of Hirokawa et al. [30]. The colonic myofibroblast cocultured-colonic epithelial cell (CM-CE) monolayers were used for analysis after four days of coculture.

### 4.2. Fluorometric Measurements in CM-CE Monolayers

The apical NHE activity was measured as described in Zhou et al. for the Caco-2BBe cells [9]. The transwell membranes with human CM-CE monolayers cultured as described above were cut off from the inserts and placed into a custom perfusion chamber with the apical side facing the objective. The chamber was fixed on the stage of an inverted microscope (Carl Zeiss AG, Oberkochen, Germany). Cells were loaded with 5 μM BCECF-AM (Life Technologies GmbH, Darmstadt, Germany) from the basolateral side. A long-distance 20× objective was used to focus on the cells through the apical side of the perfusion chamber. Assessment of NHE activity on only apical side of the colonoid monolayer was accomplished by adding Na^+^ only to the solution perfusing the apical membrane, while inhibiting basolateral NHEs by Na^+^-free perfusion buffer that contained 3 µM HOE642 to inhibit basolateral NHE1 (which may be stimulated by apical Na^+^ passing via the tight junctions) according to Zhou et al. [9]. The buffer compositions are described in Yu et al. [7]. A calibration curve (fluorescence vs. pH) was generated by incubating cells in a high potassium buffer at various pH values in the presence of the K^+^/H^+^ ionophore, nigericin. NHE inhibitors applied were 60 μM HOE642 (to inhibit NHE2 and NHE8), 3 μM HOE642 (to inhibit NHE8) and 3 μM HOE642 plus 1 µM tenapanor (to inhibit NHE3 and NHE8).

### 4.3. Immunofluorescence Imaging of Colonoids

Human 3D colonoids were prepared for immunofluorescence analysis according to Dekkers et al. [74]. Colonoid monolayer from transwell insert and myofibroblasts grown on glass cover slips were fixed with 2% and 4% PFA in PBS, respectively. The samples were then blocked and permeabilized in 5% normal goat serum (Cat. No. 10000C, Thermo Fisher Scientific GmbH, Karlsruhe, Germany) in PBS containing 0.2% Triton X-100 for 30 min at room temperature. Incubation with primary antibodies against KI67 (1:100, Cat. No. ab15580, Abcam, Berlin, Germany), MUC2 (1:200, Cat. No. sc-515032, Santa Cruz Biotechnology, Heidelberg, Germany), NHE3 (1:500, Cat. No. NBP1-82574, Novus Biologicals, Wiesbaden-Nordenstadt, Germany), NHE8 (1:500, Cat. No. PA5-106749, Invitrogen, Karlsruhe, Germany), SLC26A3 (1:200, Cat. No. sc-376187, Santa Cruz Biotechnology), SMA (1:200, Cat. No. E-AB-34268, Elabscience, Hamburg, Germany), CD90 (1:200, Cat. No. 10-652-C025, EXBIO, Vestec, Czech Republic), claudin-2 (1:200, Cat. No. 32-5600, Invitrogen, Karlsruhe, Germany) or ZO-1 (1:200, Cat. No. 61-7300, Invitrogen, Karlsruhe, Germany) was performed overnight at 4 °C in blocking solution with reduced Triton X-100 to 0.1%. After four washes in PBS containing 0.1% Triton X-100, the samples were incubated with 1:500 dilution of Alexa Fluor-conjugated secondary antibodies (Goat anti mouse 488, Cat. No. A11029, and goat anti-rabbit 568, Cat. No. A11011, Invitrogen, Karlsruhe, Germany), 2 µg/mL DAPI (Cat. No. 6843.2, Carl Roth, Karlsruhe, Germany) to stain nuclei and Phalloidin-Alexa Fluor 647 (1:500, Cat. No. ab176759, Abcam, Berlin, Germany) to stain F-actin for 1 h at room temperature, washed as mentioned above and mounted with Mowiol. For monolayers one layer and for 3D colonoids two layers of double-sided adhesive tape (Cat. No. 05338, Tesa, Norderstedt, Germany) was used as spacer between the slide and the cover slip. Samples were analyzed by a TCS SP8 confocal fluorescence microscope (LEICA Microsystems, Mannheim, Germany) with 20× multi-immersion or 63× oil immersion objectives. Image analysis was performed by LAS X (LEICA Microsystems), Fiji [75] and/or Imaris (Oxford Instruments, Abingdon, UK) software version 8.2.1.

### 4.4. Immunohistochemical Analysis of Human Colonic Biopsies

Biopsies from human transverse colon of a healthy donor were fixed in 4% PFA, washed in PBS, dehydrated and embedded in paraffin. Tissue sections with a thickness of 2 μm were prepared, deparaffinized and rehydrated. Epitope retrieval was achieved by boiling in 10 mM Tris buffer (pH 9) containing 0.5 mM EDTA. The samples were quenched in 50 mM NH_4_Cl in TBS, blocked in 3% BSA and exposed overnight at 4 °C to anti-SLC26A3 antibody (1:200, Cat. No. sc-376187, Santa Cruz Biotechnology) prepared in PBS containing 1% BSA and 0.1% Triton X-100. After washing in PBS, the samples were incubated with Goat anti mouse 488 secondary antibody (1:500, Cat. No. A11029, Invitrogen, Karlsruhe, Germany) and 2 µg/mL DAPI (Cat. No. 6843.2, Carl Roth, Karlsruhe, Germany) for 1 h at room temperature and then washed in PBS. Finally, the sections were mounted in Fluoromount-G (SouthernBiotech, Birmingham, AL, USA) and images were acquired using Olympus FV1000 confocal microscope (Evident Europe GmbH, Hamburg, Germany).

### 4.5. RT-qPCR Analysis

Total RNA was prepared using RNeasy Mini Kit (Cat. No. 74106, Qiagen GmbH, Hilden, Germany). Reverse transcription with oligo dT primers was performed using RevertAid First Strand cDNA Synthesis Kit (Cat. No. K1622, Thermo Fisher Scientific GmbH, Karlsruhe, Germany). Quantitative PCR was performed with a Rotor Gene Q system (Qiagen) using 10 ng of total cDNA, 500 nM of each forward and reverse primers (Appendix A) and qPCRBIO SyGreen Mix Lo-ROX (Cat. No. PB20.11-51, Nippon Genetics, Dueren, Germany).

### 4.6. SDS-PAGE and Western Blot

Samples were solubilized in a cold lysis buffer composed of 60 mM HEPES, 140 mM NaCl, 1% Triton X-100, 0.5% sodium deoxycholate, 1 mM EDTA and 1 mM EGTA supplemented with protease inhibitor cocktail (Cat. No. 4693159001, Roche, Mannheim, Germany). Debris were sedimented by centrifuging at 2000× *g* for 5 min at 4 °C. Then, 50 µg of each lysate (BCA assay, Cat. No. 23225, Thermo Fisher Scientific GmbH, Karlsruhe, Germany) was resolved by SDS-PAGE and wet-transferred to PVDF membrane (Cat. No. 10600023, Cytiva, Freiburg im Breisgau, Germany). After blocking in TTBS containing 5% nonfat milk, samples were probed with primary antibodies against beta-actin (1:2000, Cat. No. 3700, Cell Signaling Technology, Leiden, The Netherlands), SLC26A3 (1:500, Cat. No. sc-376187, Santa Cruz Biotechnology), or NHE3 (1:500, Cat. No. NBP1-82574, Novus Biologicals, Wiesbaden-Nordenstadt, Germany) prepared in 1% BSA-TTBS solution overnight at 4 °C. The membranes were washed with TTBS, exposed to HRP-conjugated secondary antibodies (1:10,000, Cat. No. G-21040 or G-21234, Invitrogen, Karlsruhe, Germany), washed and developed with ECL solution (Cat. No. RPN2209, Cytiva). The signal was detected and analyzed with a Fusion FX device equipped with Fusion-Capt Advance software.

### 4.7. Statistics

Microsoft Excel 2016 and GraphPad Prism version 8.0.2 (GraphPad Software Inc., San Diego, CA, USA) were used for data analysis. Data are presented as mean ± SEM. Statistical significance as determined by *p*-values from unpaired, two-tailed parametric *t*-test is shown as * *p* < 0.05, ** *p* < 0.01 or *** *p* < 0.001.

## Figures and Tables

**Figure 1 ijms-24-04266-f001:**
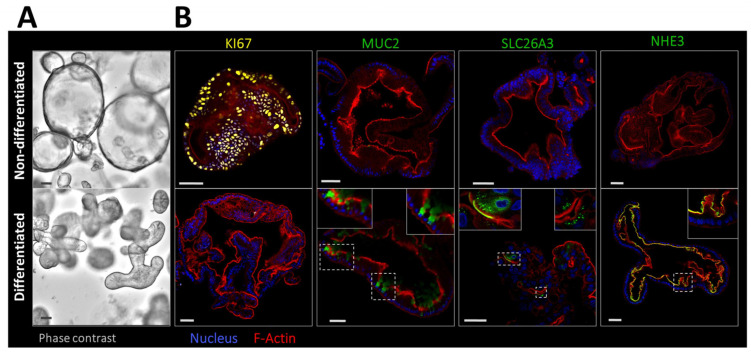
Colonoids generated from crypts isolated from human transverse colon and cultured in nondifferentiating (expansion) medium and in differentiating medium lacking Wnt signaling stimuli. (**A**) The morphological features of 3D colonoids at day 4 in the respective expansion or differentiation medium. (**B**) IHC staining of whole mount organoids with many proliferative (KI67-positive) cells when cultured in expansion medium (upper panels), and with differentiation markers MUC2, NHE3 and SLC26A3 when cultured in differentiation medium (lower panels). Scale bar 50 µm. The dotted squares indicate the areas whose magnification is displayed in the upper right and left corners of the respective micrographs.

**Figure 2 ijms-24-04266-f002:**
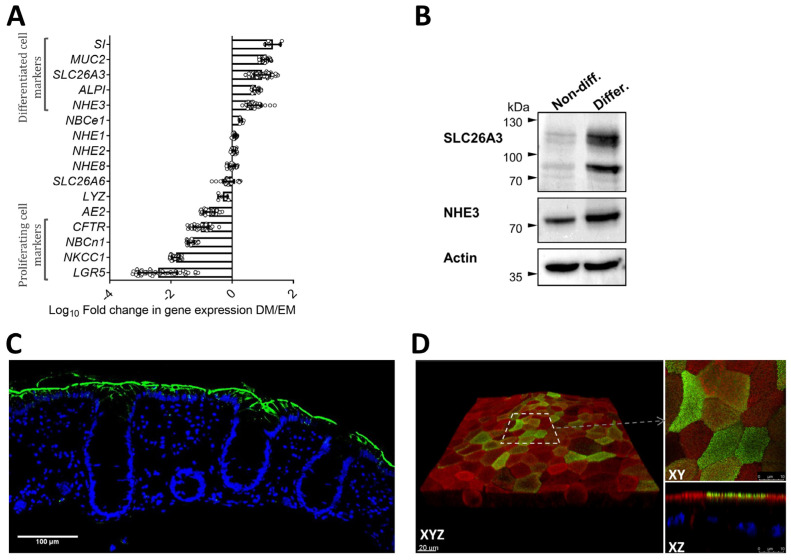
Expression of ion transporters in human colonoid cultures and colon tissue. (**A**) Log_10_ fold change in mRNA expression levels of a panel of genes (mostly coding for ion transport proteins) between the nondifferentiated state (3D in expansion medium for several days after the removal of Y-27632 and CHIR-99021), representing cryptal base and lower neck cells, and the differentiated state, representing the cells in the cryptal mouth/surface. n ≥ 5 separate cultures. Genes that assemble close to zero in this graph likely have a homogenous expression pattern along the cryptal axis. (**B**) Western blot analysis of the absorptive enterocyte markers SLC26A3 and NHE3 in nondifferentiated and differentiated 3D colonoids. (**C**) Formalin-fixed, paraffin-embedded (FFPE) section of human transverse colon stained for SLC26A3 (green) and nuclei (blue). SLC26A3 is strongly expressed on the luminal side of mucosal cells from the surface and crypt neck regions, where colonic epithelium is terminally differentiated. (**D**) Expression of SLC26A3 in human colonoid monolayer after four days of differentiation. 3D reconstruction of a region of monolayer (left panel) shows that SLC26A3 is expressed at different levels in different cells. Right panel shows top view and cross-section of the selected area in higher magnification where SLC26A3 is apically localized. Green: SLC26A3, red: F-actin, blue: nuclei (excluded in XYZ).

**Figure 3 ijms-24-04266-f003:**
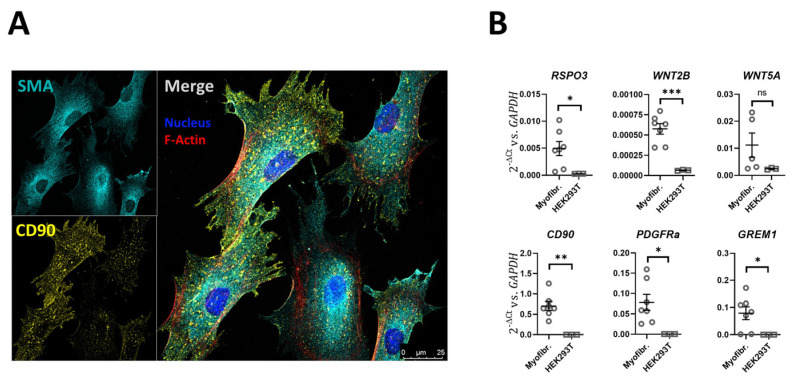
Morphological features and mRNA expression panel of myofibroblast cultured from colonic lamina propria. (**A**) Immunohistochemical staining for smooth-muscle actin (SMA), CD90, phalloidin (F-actin) and DAPI (nuclei). (**B**) Myofibroblast (Myofibr.) mRNA expression of a panel of genes encoding for proliferative and differentiation factors in the myofibroblasts, in comparison to HEK293T cells. n = 5–7 different myofibroblast cultures, *t*-test, * *p* < 0.05, ** *p* < 0.01, *** *p* < 0.001.

**Figure 4 ijms-24-04266-f004:**
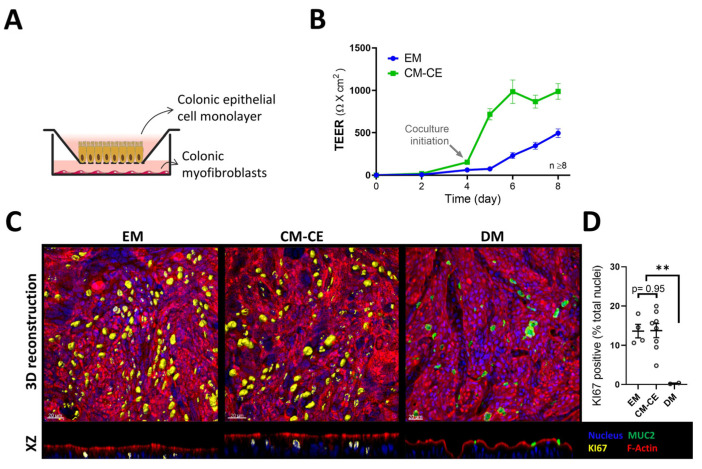
Colonic myofibroblast (CM)–colonic epithelial cells (CE) coculture system. (**A**) Schematic diagram of the CM-CE transwell monolayer cultures. (**B**) Development of the transepithelial electrical resistance (TEER) during the days of the monolayers in culture. In the subconfluent phase, the colonoid cells were cultured in “expansion” medium, the myofibroblasts were plated onto the bottom of transwell plates as described in material and methods. After confluency, the transwell cups were placed in wells containing myofibroblast monolayers at the bottom (coculture) or in wells containing expansion medium without myofibroblasts. TEER increased much faster in the CM-CE cocultures. (**C**) IHC for KI67 (surface rendering in 3D reconstructions for facilitated counting of KI67-positive nuclei (yellow) and MUC2 positive mucin granules (green) in 2D organoid cultures, EM: cultured for 4 days after confluency in expansion medium, CM-CE: 4 days coculture, DM: 4 days in differentiation medium. Ratio of KI67 positive nuclei versus total nuclei was quantified and illustrated in (**D**). *t*-test, ** *p* < 0.01.

**Figure 5 ijms-24-04266-f005:**
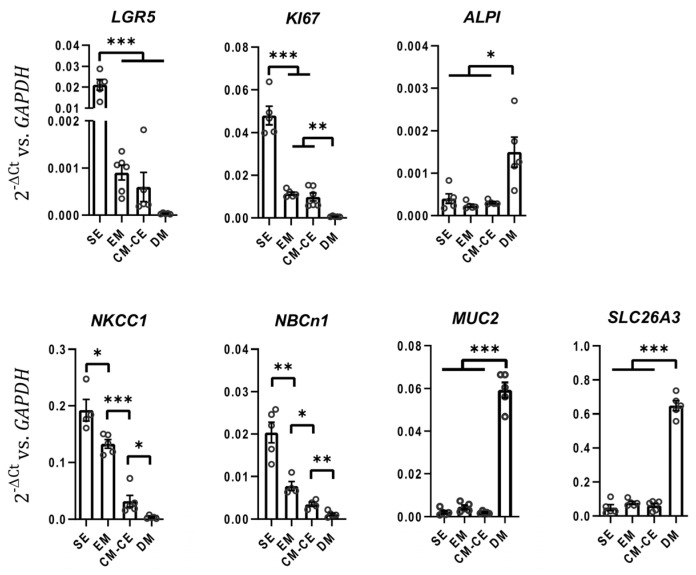
Different stages of colonoid cultures represent different regions of the colonic crypts in gene expression profile. Relative mRNA expression of a panel of proliferation/differentiation markers in addition to ion transporters with differential spatial expression along cryptal axis is shown in stem cell enriched 3D colonoids (SE) and non-differentiated (EM), CM-CE and differentiated (DM) colonoid monolayers; *t*-test, * *p* < 0.05, ** *p* < 0.01, *** *p* < 0.001.

**Figure 6 ijms-24-04266-f006:**
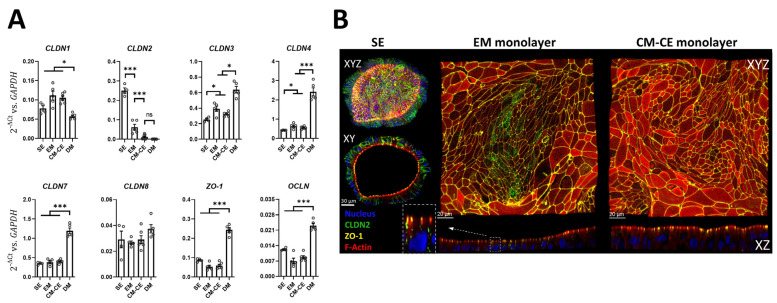
Gene expression for a panel of tight junction proteins in different stages of colonoid culture. (**A**) Colonoids grown as stem cell enriched 3D organoids (SE), as monolayers in expansion medium (EM), as CM-CE cocultures, and as differentiated monolayers (DM) for four days. The conspicuous difference between EM and CM-CE monolayers is the downregulation of *claudin-2*. *t*-test, * *p* < 0.05, *** *p* < 0.001. (**B**) IHC of SE, EM, and CM-CE monolayers, displaying the dramatic downregulation of claudin-2 (Green: CLDN2, blue: nucleus, yellow: ZO-1, red: F-actin).

**Figure 7 ijms-24-04266-f007:**
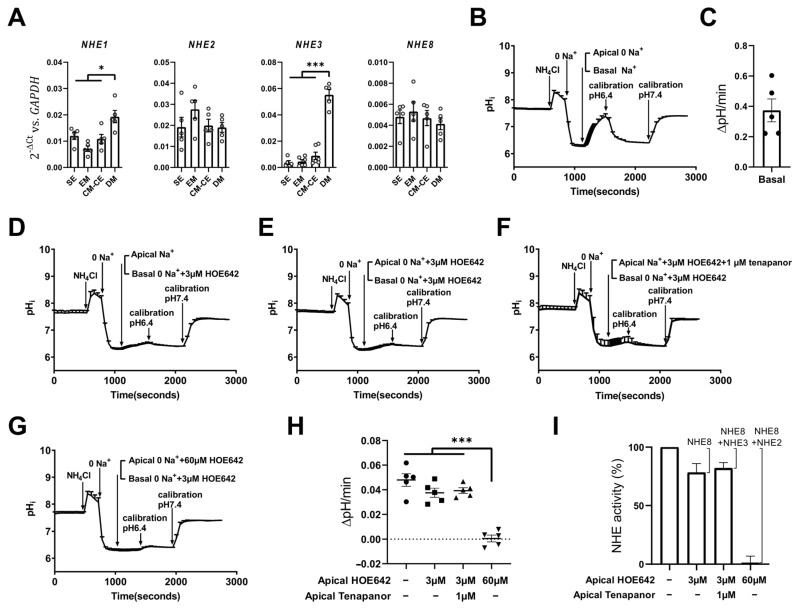
NHE expression and activity in CM-CE monolayers. (**A**) The mRNA expression levels of *NHE* isoforms in SE 3D organoids and EM, CM-CE and DM monolayers, characterized in Figure 4, Figure 5 and Figure 6. (**B**,**C**) Basolateral NHE activity analyzed fluorometrically in the CM-CE monolayers by removal and re-addition of Na^+^ to the basolateral perfusate. (**D**–**I**) Apical NHE activity in the CM-CE monolayers measured by removal and re-addition of Na^+^ to the apical perfusate in the absence or presence of NHE-specific inhibition. For apical NHE activity measurements, the basolateral NHE1 was inhibited by removal of Na^+^ and addition of 3 µM HOE642 in the basolateral perfusate. Representative pH_i_ traces for apical NHE activity in control (**D**), with 3 μM HOE642 to inhibit NHE8 (**E**), with 3 μM HOE642 plus 1 μM tenapanor to inhibit NHE3 and NHE8 (**F**), and with 60 μM HOE642 to inhibit NHE8 and NHE2 (**G**) conditions are illustrated. (**H**) The pH_i_ recovery rates shown as ΔpH/min without and with the different inhibitor concentrations. (**I**) Method of estimating the respective NHE isoform activities: Na^+^-dependent pH_i_ recovery rate in the absence of inhibitors was set to 100%, and by subtracting the respective percentage rates in the presence of the respective inhibitor concentrations from the total apical activity (100%), the relative NHE-specific activities were calculated. The calculated exact values in percentage of total apical NHE activity are as follows: NHE2 82 ± 10.7; NHE8 21.7 ± 17.2; NHE3 1.21 ± 12.9 (each dot represents one individual CM-CE monolayer, n = 5, mean ± SEM, *t*-test, * *p* < 0.05, *** *p* < 0.001).

## Data Availability

Not applicable.

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
