# Peer review of "Human Colonoid–Myofibroblast Coculture for Study of Apical Na+/H+ Exchangers of the Lower Cryptal Neck Region"

_ijms, 2023, doi:10.3390/ijms24054266_

Round 1

Reviewer 1 Report (Previous Reviewer 1)

This manuscript can be accepted in its current version. There are no further comments from my position.

Author Response

Thank you very much for reviewing our manuscript.

Reviewer 2 Report (New Reviewer)

The authors have constructed colonoid-myofibroblast cocultures using which they’ve studied the ion exchangers. It is a nice work with an interesting topic and informative results. I only have a few minor comments for the authors to improve their manuscript.

-        The work is novel; however, this novelty is not clearly mentioned in the introduction. Please consider emphasizing it.

-        I suggest better labeling the figure panels. Currently, there are multiple panels with no labels. For example, see Figures 1, 3, 4, and 5.

-        I did not find discussing some Supplementary figures in the main text. They need to be mentioned at least once. Also, Supplementary Figure 3 is very informative, and so I suggest moving it to the main text.   

I suggest publishing this work in this journal, after authors revise it based on the mentioned comments. 

Author Response

Point by point response to reviewer 2: Please note: For better comprehension, our answers are pasted directly below the reviewers questions.

Comments and Suggestions for Authors

The authors have constructed colonoid-myofibroblast cocultures using which they’ve studied the ion exchangers. It is a nice work with an interesting topic and informative results. I only have a few minor comments for the authors to improve their manuscript.

-        The work is novel; however, this novelty is not clearly mentioned in the introduction. Please consider emphasizing it.

Answer: Thank you for the comment. We have now emphasized 1. The need to create model systems that allow the study of ion transporters in native human intestinal epithelial cells, as opposed to cell culture system derived from malignant tumours, and 2. the novelty of the work being the coculture system, which permitted a modeling of the lower cryptal region in the human colon, with an accessible apical membrane for functional study. And indeed, the results differ from those obtained in Caco2bbe and in differentiated HT29 cells, which are also very useful models. But their cellular fate is determined, even in the preconfluent stage. They do not display the full range of differentiation from nondifferentiated, stem cell rich and proliferating epithelia with a high expression of components of the anion secretory machinery, such as CFTR, NKCC1, Claudin-2 etc., to epithelia with a strong decrease in the expression of the aforementioned, paralleled by a strong increase of absorptive and/or mucus and endocrine secretory cells.

-        I suggest better labeling the figure panels. Currently, there are multiple panels with no labels. For example, see Figures 1, 3, 4, and 5.

Answer: Done, thank you.

-        I did not find discussing some Supplementary figures in the main text. They need to be mentioned at least once. Also, Supplementary Figure 3 is very informative, and so I suggest moving it to the main text.  

Answer: Done, thank you. We moved the contents of supplementary Figure 3 to the Figure 2, where it fits best.

Reviewer 3 Report (New Reviewer)

The authors have developed a new in vitro model of colonic lower crypt cells in which the apical membrane is accessible to study Na+/H+ exchangers. Primary epithelial cells are co-cultured with primary myofibroblasts, isolated from biopsies. Myofibroblasts grown as monolayers on the bottom of transwells, with colonic epithelial cells grown on filters above.

Major points

The main finding is that the NHE2 isoform is the predominant apical Na+/H+ exchanger. However, it is difficult to understand the key importance of this finding. The introduction sets out that NHE2 and NHE8 isoforms are believed to have important but fundamentally different roles in cellular physiology. The article would benefit from a more clear explanation of:

(1)    What can we already conclude from these observations that NHE2 is predominant over NHE3 in this system?

(2)    What will be the most important follow-on studies that this co-culture system now makes possible?

Minor points

Conclusion, lines 283-285: This section is difficult to understand because the sentences:

“However, as discussed in those reviews, important questions remain open.

One of those questions is the expression and function of NHE2 and NHE8 in native human colonocytes?”

do not say what the questions are. Are the questions: (1) is NHE2 expressed in native human colonocytes? and (2) is NHE8 expressed in native human colonocytes? Or is the question: what are the relative expression levels of NHE2 and NHE8 in human colonocytes? Or is the question something about which of the two isoforms regulates some aspect of cell physiology in native human colonocytes? Or is it something else entirely? The paper would be improved if the authors could explain this more clearly.

It is not clear how the observations of changes in transepithelial resistance (TEER), relate to the function of the NHE2 and NHE8 isoform Na+/H+ exchangers. The paper would be improved if the authors could explain this more clearly.

Overall, this reviewer found the conclusion section from difficult follow. The narrative seemed to jump from one topic to another without showing how they related to each other or to the main research finding of the paper, which was establishment of a new co-culture cell system. The paper would be improved if the authors could explain more clearly how the discussion topics relate to each other and to the results obtained in this new co-culture system.

Author Response

Point by point response to Reviewer 3: Please note: For better comprehension, our answers are pasted directly below the reviewers questions.

The authors have developed a new in vitro model of colonic lower crypt cells in which the apical membrane is accessible to study Na+/H+ exchangers. Primary epithelial cells are co-cultured with primary myofibroblasts, isolated from biopsies. Myofibroblasts grown as monolayers on the bottom of transwells, with colonic epithelial cells grown on filters above.

Major points

The main finding is that the NHE2 isoform is the predominant apical Na+/H+ exchanger. However, it is difficult to understand the key importance of this finding. The introduction sets out that NHE2 and NHE8 isoforms are believed to have important but fundamentally different roles in cellular physiology. The article would benefit from a more clear explanation of:

  • What can we already conclude from these observations that NHE2 is predominant over NHE3 in this system?
  • What will be the most important follow-on studies that this co-culture system now makes possible?

Answer: Thank you for the suggestions. We have now defined the different topics of the discussion before we plunge into DISCUSSING them. We also added the answer to the above questions.

Minor points

Conclusion, lines 283-285: This section is difficult to understand because the sentences:

“However, as discussed in those reviews, important questions remain open.

One of those questions is the expression and function of NHE2 and NHE8 in native human colonocytes?”

do not say what the questions are. Are the questions: (1) is NHE2 expressed in native human colonocytes? and (2) is NHE8 expressed in native human colonocytes? Or is the question: what are the relative expression levels of NHE2 and NHE8 in human colonocytes? Or is the question something about which of the two isoforms regulates some aspect of cell physiology in native human colonocytes? Or is it something else entirely? The paper would be improved if the authors could explain this more clearly.

Answer: This section has been rewritten, and the part regarding NHE8 was shortened, because our study does not allow to say much about this isoform, given it low functional activity in the model that we studied.

It is not clear how the observations of changes in transepithelial resistance (TEER), relate to the function of the NHE2 and NHE8 isoform Na+/H+ exchangers. The paper would be improved if the authors could explain this more clearly.

Answer: We have now outlined the fact that we discuss two separate issues: Firstly, the necessary information regarding the coculture system are discussed, then, the strategy and the results of the NHE experiments are discussed. Finally, an outlook is given on the potential future investigations that can follow, using this coculture system.

Overall, this reviewer found the conclusion section from difficult follow. The narrative seemed to jump from one topic to another without showing how they related to each other or to the main research finding of the paper, which was establishment of a new co-culture cell system. The paper would be improved if the authors could explain more clearly how the discussion topics relate to each other and to the results obtained in this new co-culture system.

This manuscript is a resubmission of an earlier submission. The following is a list of the peer review reports and author responses from that submission.

Round 1

Reviewer 1 Report

In this manuscript, the authors characterized human 3D and 2D intestinal epithelial organoids using different media conditions and during a myofibroblasts coculture system. They analyzed the intestinal barrier integrity, tight junction expression as well as cell differentiation and Na+/H+ exchange.

The authors present an interesting study, which is mostly a pleasure to read. As the authors are predominantly interested on Na+/H+ exchanger, this part of the manuscript is difficult to understand. The results for the Na+/H+ exchangers were described very briefly and the results become a little more understandable once the figure legend is also read. It is not clear to me how Fig. 7H shows that NHE2 shows activity for 90% of the pHi recovery, NHE8 for 10% and NHE3 for none. I would be very grateful for a revision of this part of the results. Furthermore, the discussion would also benefit from a revision and a focus.

Reviewer 2 Report

The manuscript describes a co-culture method for primary human colonic epithelium that models the characteristics of the transit-amplifying region of the intestinal crypt.  This fits a gap between known culture methods for undifferentiated (stem-like) and fully differentiated (villus/surface) colonic epithelium.  Using this technique, studies of intestinal Na/H exchangers confirm in human colonic crypt epithelium that NHE2 provides the dominant Na/H exchanger in colonic crypts, as has previously been documented in mice.  The manuscript is well-written, but a few comments are warranted:

Major

1.     A major point of the manuscript is that the myofibroblast co-culture method provides a means to model the transit-amplifying zone of the human colonic crypt, as pointed out in the text (Abstract, line 27, Results, line 158, Discussion, line 413).  Previous studies in mouse enteroid monolayers counted cells that stained for EdU but not for the stem cell marker Lgr5 to identify the transit-amplifying cells (Dev Cell 56: 356-65, 2021).  It would greatly increase the impact of the manuscript if it were shown that the Ki67 cells do or do not colocalize with LGR5 by immunofluorescence in the co-culture monolayers.  Otherwise, it may be that the co-culture soluble factors just somehow increased the tight junctions between stem and differentiating cells

Minor

1.     Figures 2 and 3 are cut off at the right margin.

2.     Please indicate the significance of CD90 staining for the myofibroblasts in the text.

3.     There is a degree of loose definition in that it is not always clear when the text refers to the crypt base versus the transit-amplifying zone, which is not at the crypt base (crypt mid-region?).  For example, line 30 of the Abstract is referring to the method proposed to model the transit-amplifying zone “at the base of the colonic crypts”.  There are several other instances in the rest of the manuscript.

4.     It would be helpful to the reader if the definition and its abbreviation were more clearly defined in the text at the beginning of the manuscript, for example in the last paragraph of the Introduction.  The reader does not need to jump to the Methods to find the abbreviations used in the Results.

5.     Figure 1:  It is interesting that DRA staining is patchy in the colonoids (Fig. 1) but intensely express in the Western blot.  Is there an explanation?

6.     Figure 2A:  It might be helpful to put an upward arrow to left labeled with “Differentiation” to improve the ease of understanding the graph.

7.     There is a formatting error in that many words are hyphenated in mid-sentence.

8.     Both the Introduction and the Discussion are too long.  In parts, they seem more like a review article or dissertation, for example, reviewing the past and speculating on the development of a new culture method for NHE8.  Please shorten these sections to be more succinct.

2.